Why do leaf-tying caterpillars abandon their leaf ties?

Sliwinski Michelle
Sigmon Elisha elisha.sigmon@gmail.com
Department of Biological Sciences, George Washington University , Washington, DC , USA
Sanders Nathan
Electronic publication date: 2013 Sep 26
Publication date: 2013
Volume: 1
Electronic Location ID: e173
Received 2013 Jul 25; Accepted 2013 Sep 9
Copyright: © 2013 Sliwinski and Sigmon
Copyright year: 2013
Copyright holder: Sliwinski and Sigmon
License: This is an open access article distributed under the terms of the Creative Commons Attribution License, which permits unrestricted use, distribution, and reproduction in any medium, provided the original author and source are credited.
License URL: https://creativecommons.org/licenses/by/3.0/

Keywords: Caterpillar, Ecosystem engineer, Resource requirements, Leaf shelter, Development, Site fidelity

Funding: George Washington University’s Luther Rice Fellowship Program National Science Foundation DEB-1210600 Funding was provided by George Washington University’s Luther Rice Fellowship Program to MS and the National Science Foundation (DEB-1210600) to ES. The funders had no role in study design, data collection and analysis, decision to publish, or preparation of the manuscript.

==============================
Leaf-tying caterpillars act as ecosystem engineers by building shelters between overlapping leaves, which are inhabited by other arthropods. Leaf-tiers have been observed to leave their ties and create new shelters (and thus additional microhabitats), but the ecological factors affecting shelter fidelity are poorly known. For this study, we explored the effects of resource limitation and occupant density on shelter fidelity and assessed the consequences of shelter abandonment. We first quantified the area of leaf material required for a caterpillar to fully develop for two of the most common leaf-tiers that feed on white oak, Quercus alba. On average, Psilocorsis spp. caterpillars consumed 21.65 ± 0.67 cm2 leaf material to complete development. We also measured the area of natural leaf ties found in a Maryland forest, to determine the distribution of resources available to caterpillars in situ. Of 158 natural leaf ties examined, 47% were too small to sustain an average Psilocorsis spp. caterpillar for the entirety of its development. We also manipulated caterpillar densities within experimental ties on potted trees to determine the effects of cohabitants on the likelihood of a caterpillar to leave its tie. We placed 1, 2, or 4 caterpillars in ties of a standard size and monitored the caterpillars twice daily to track their movement. In ties with more than one occupant, caterpillars showed a significantly greater propensity to leave their tie, and left sooner and at a faster rate than those in ties as single occupants. To understand the consequences of leaf tie abandonment, we observed caterpillars searching a tree for a site to build a shelter in the field. This is a risky behavior, as 17% of the caterpillars observed died while searching for a shelter site. Caterpillars that successfully built a shelter traveled 110 ± 20 cm and took 28 ± 7 min to find a suitable site to build a shelter. In conclusion, leaf-tying caterpillars must frequently abandon their leaf tie due to food limitation and interactions with other caterpillars, but this is a costly behavior.

Introduction

Leaf-tying caterpillars are essential components of forest ecosystems due to their role as physical ecosystem engineers—organisms that create or modify their habitat and in turn affect the community (Jones, Lawton & Shachak, 1997). Caterpillars pull together overlapping leaves and fasten them together with silk to create leaf ties, a type of shelter that serves as room and board for the developing insects (Lill & Marquis, 2007). On white oak trees, these ties are secondarily colonized by other leaf-tying caterpillars as well as over 100 species of arthropods, including aphids, psocids, Asiatic oak weevils, rove beetles, thrips, and collembolans (Lill & Marquis, 2004; Sigmon & Lill, 2013). Leaf ties offer many potential advantages to residents, including protection from predators and harsh climatic conditions (Lill & Marquis, 2007), and the presence of leaf ties on a tree has been shown experimentally to increase arthropod abundance and species richness (Lill & Marquis, 2003; Lill & Marquis, 2004; Wang, Marquis & Baer, 2012).

Leaf ties are formed by caterpillars that have been oviposited between overlapping leaves or have abandoned a previous tie. When caterpillars build additional ties, they create a new habitat and increase their engineering effect. However, the specific reason(s) why caterpillars leave their shelters prior to completing development has not been examined. Many shelter-building caterpillars have been observed to physically compete over their shelters by using territorial vibratory signals or physically aggressive behavior, at times pushing one another to either gain access to or defend leaf shelters (Berenbaum, Green & Zangerl, 1993; Yack, Smith & Weatherhead, 2001; Scott et al., 2010; E Sigmon, unpublished data). This suggests that competition over existing shelters is common and that shelters, or sites to build shelters, are a limiting resource. Since leaf-tying caterpillars generally do not feed outside of their shelter (Lill & Marquis, 2004), we assume that competition between leaf tiers is based on the need for food resources and space inside of the tie. If this is the case, then when resources are low, a caterpillar should leave or be forced to find another shelter within which to feed. However, previous studies have shown that new, unoccupied ties and damaged, occupied ties are equally attractive to colonizing herbivores (Lill, 2004; Lill & Marquis, 2004). This is despite the fact that caterpillars that are reared in groups within a single shelter achieve a lower pupal mass than those reared individually (Lill et al., 2007). This suggests that food resources are not the only limiting factor for site selection. Leaf-tiers may be further limited by available sites for creating leaf ties (Marquis & Lill, 2010), forcing them to colonize occupied leaf ties if there are no suitable sites for building new ties; this is especially likely later in the season when most suitable sites have been used by previous generations of herbivores.

While movement of leaf-tying caterpillars between ties on white oak has been observed (Lill & Marquis, 2004), the frequency of this behavior and the motivations behind this movement are currently unknown. This behavior is expected to be extremely perilous for caterpillars, as they risk attack from predators, dislodgement from the tree, and exposure to harsh climatic conditions. However, the movement of caterpillars and the establishment of additional leaf ties will increase the engineering effect due to the creation of new microhabitats that can be used by other arthropods. If common, these bouts of caterpillar movement are predicted to increase habitat heterogeneity and biodiversity on a plant.

The focus of this study was to examine why leaf-tying caterpillars move among ties during the course of larval development. The feeding ecology and behavior of the most common species of leaf-tying caterpillars found on white oak in the eastern United States were examined. Interactions among caterpillars within leaf ties are common and are expected to determine the “residence time” of occupants within their shelters, particularly when shelter size places constraints on food availability. However, it is not well understood whether caterpillars leave shelters only after food resources are depleted or whether behavioral interactions among co-occurring caterpillars anticipate food-limitation and drive the “losers” of aggressive contests from the shelters prior to food resource depletion. Additionally, the consequences of shelter abandonment and the process of finding a new leaf tie site have not been investigated. If the risks associated with abandoning a shelter are high, then caterpillars may share a leaf tie despite food limitations or territorial encounters. This study explored the role that food limitation, competitive interactions, and risks of leaving a leaf tie play in shelter abandonment through a series of experiments: (1) We determined the amount of food resources consumed by the most common leaf-tying caterpillars on white oak to complete development; (2) we compared these resource requirements to the size of naturally occurring leaf ties to determine if food limitation is common; (3) we manipulated caterpillar densities within experimental leaf ties to determine the effects of interactions between caterpillars on their propensity to abandon their ties; and (4) we observed caterpillars searching for a site to build a leaf tie and documented their behaviors. Together, these experiments helped to elucidate the dynamic nature of species interactions that characterize the dominant players in the oak leaf tier community.

Materials and Methods

Experiment 1. Feeding ecology

The objective of the first experiment was to quantify the amount of plant material required for a leaf-tying caterpillar to fully develop, in order to determine what role resource limitation plays in shelter abandonment. We obtained eggs of two of the most common leaf-tying caterpillars, Psilocorsis quercicella and P. cryptolechiella, from Little Bennett Regional Park (LBRP), Montgomery Co., MD, and existing laboratory colonies. Nine of the 21 P. cryptolechiella caterpillars were from the lab colony from 3 families; all other caterpillars were field collected. Lab-bred and field-collected P. cryptolechiella caterpillars did not differ in the area of leaf eaten, pupal mass or development time (all P > 0.25). Experimental leaf ties were created by clipping two overlapping white oak (Quercus alba) leaves together with a metal hair clip, to provide an area to build a shelter. Initially, P. quercicella and P. cryptolechiella hatchlings were placed in experimental leaf ties enclosed in mesh bags on potted white oak trees in a shade house at George Washington University. For the second generation, each caterpillar was kept in an experimental leaf tie in a large deli container, with the stems of the leaves in aquapics filled with water to keep the leaves fresh, with leaves replaced weekly. Lill et al. (2007) compared survivorship, pupal mass, and development time of caterpillars forced to constantly rebuild shelters to those that were not disturbed and found no difference. The caterpillars were kept in a growth chamber at 18:27°C with a 12:12 h light cycle.

The caterpillars were examined periodically to monitor growth, and were removed from the experimental leaf tie when they pupated. The pupae were weighed to the nearest 0.01 mg and the sex was determined using a dissecting scope. Leaves from each experiment were removed from the tree or container, cleaned, dried, and pressed. The leaf damage (skeletonized area in cm2) was traced on transparency paper and then scanned as images onto the computer. The area of leaf consumed was quantified using the ImageJ® software program (Rasband, 2012). In total, 27 P. quercicella and 21 P. cryptolechiella were reared completely from egg hatch to pupation.

We calculated a correlation matrix for three performance measures: area of plant material consumed, development time, and pupal mass. We performed three analyses of variance tests (ANOVA) to test the effects of species, sex, and the species × sex interaction on the area eaten, development time, and pupal mass of caterpillars, using the other performance measures not being tested as covariates. Non-significant covariates and interactions were dropped from the final models. All analyses were performed in R version 2.15.2 (R Core Development Team, 2012).

Experiment 2. Area of natural ties

The average size of natural leaf ties collected from LBRP was calculated to determine the amount of resources available to caterpillars in situ. By comparing resource requirements in the lab to the size of natural ties in the field, we determined whether natural ties are sufficient to sustain a single caterpillar or if resource limitation will force a caterpillar to leave the tie and build a new one elsewhere on the plant.

Naturally occurring leaf ties were collected from white oak trees in LBRP during midsummer 2012 when leaf ties were abundant. Thirty ties were collected on June 7th, 33 ties were collected on June 27th, 31 ties were collected on August 2nd, and 64 ties were collected on August 23rd, for a total of 158 ties. For each tie, the area of overlap between leaves, representing the area of the natural tie (Fig. 1A), was traced on tracing paper while the tie was intact. The area of overlap of natural leaf ties was calculated in square centimeters using the ImageJ® software program (Rasband, 2012). Since caterpillars feed upon both leaf surfaces, the area of the tie was doubled to represent the surface area of leaf available for consumption by the occupants for comparison to Experiment 1. In addition to calculating the area of the natural ties, the abundance and species of caterpillars found inside each tie was noted. Small caterpillars were lab-reared until they could be identified. Due to prior work in this system, we were able to determine the species that built some of the ties due to species-specific patterns of shelter construction and size of caterpillars found in the tie.

Figure 1 Drawing of (A) a leaf tie that shows two overlapping white oak leaves and the characteristic shelter shapes made by (B) Psilocorsis cryptolechiella/reflexella and (C) Psilocorsis quercicella.

(A) The grey region comprises the area of the leaf tie. The area of this region was doubled to represent the area of both leaf surfaces that is available for consumption by caterpillars.

The resource availability (doubled area) of the natural ties was compared with the resource requirements of each species obtained from Experiment 1: Feeding Ecology to determine whether limited resources have a likely role in shelter use and abandonment. Additionally, an ANOVA was performed to test the effect of the number of occupants (0, 1, 2, and 3 or more) within a tie on the area of the tie. We also performed an ANOVA to test the effect of the species of the original builder on the size of natural ties, for those that could be determined (n = 71). To account for multiple tests we used a Bonferroni correction. When results were significant, post-hoc tests were performed using Tukey’s HSD Test.

Experiment 3. Density effects on caterpillar movement

The objective of this experiment was to determine how the density of leaf-tie occupants influences the likelihood that resident caterpillars will leave their leaf ties. This experiment was conducted on potted white oak trees both in the shade-house and in the laboratory. Caterpillars were either field-collected or obtained from laboratory colonies. Overlapping leaves were clipped together using metal hair clips to create experimental ties that were approximately 12–14 cm2 in size, which preliminary data suggested was the average size of natural leaf ties. Each trial consisted of three density treatments with 1, 2, or 4 late-instar caterpillars in a tie of the same species, either P. quercicella or P. cryptolechiella. Initial caterpillar mass did not differ between density treatments (F2,54 = 0.09, p = 0.9141). “Bait ties” were set up by clipping overlapping leaves together 20 cm, 40 cm and 60 cm from the original experimental tie to control for plant architecture (i.e., the availability of places to build ties) to ensure that caterpillars did not remain in the tie due to a lack of nearby sites for building ties. A mesh bag was placed over each tie for 2 h in order to force the caterpillars to begin interacting before they had a chance to leave the tie. The mesh bags were then removed and the caterpillars were censused every morning and afternoon for a total of 11 census periods (5 1/2 days), recording all movements. We were unable to mark individual caterpillars, thus we noted the number of caterpillars that remained in the tie instead of the fate of each caterpillar. For caterpillars that left their shelters, we searched the tree for new shelters and measured the shortest distance along tree branches between its starting shelter and the new shelter it built and noted whether they built new ties or occupied “bait ties”. Since there were some observations of predation by spiders and adverse weather conditions on the roof, the last three trials were performed on potted trees in the laboratory to minimize effects from the outside disturbances. We performed 20 trials; 12 trials with P. quercicella and 8 with P. cryptolechiella/reflexella. Psilocorsis cryptolechiella and P. reflexella are indistinguishable in early instars, so the species of field-collected caterpillars could not be definitively determined prior to the experiment. These two species behave similarly in territorial competition (E Sigmon, unpublished data), so these two species were combined for analysis.

An ANOVA was used to test the effects of density treatment and species on the census period when the first caterpillar left the tie. An analysis of covariance (ANCOVA) was conducted to compare the proportion of the original tie occupants remaining in the tie among census periods, density treatments and species. Thus the response for individual caterpillars is 0 or 1 while the response for ties with 4 caterpillars can be 0, 0.25, 0.5, 0.75, or 1. We recognize the limitations of this response variable, but were limited by experiment design and the inability to follow individual caterpillars. When treatment effects of ANOVAs were significant, post-hoc tests were performed using Tukey’s HSD Test. We also used the Fischer Exact Probability Test to examine if there was a difference between density treatments in whether caterpillars left their ties overnight or during the day.

Experiment 4. Searching behavior

We know that leaf-tying caterpillars frequently move between ties, but they have never been observed outside of a leaf tie. We monitored caterpillars searching for a site to build a leaf tie to explore how they find a suitable site, how long this takes, and the dangers associated with being out of a tie. Observations were made on understory white oak trees at LBRP. Individual late-instar Psilocorsis quercicella caterpillars (N = 29) were placed on a branching point at the beginning of the observation. Caterpillars were watched moving throughout the tree until they began to attach silk to leaves to build a shelter. We used a stopwatch to record the time the caterpillar spent crawling and resting (not moving) to the nearest 1 min. We also tallied the number of times the caterpillar walked on a new leaf, walked on a leaf it had been on before, stopped to rest, fed, and fell off the branch. We followed the path of each caterpillar and measured this path to the nearest 1 cm with a string to determine how far the caterpillar traveled from the start of the observation to where it built a shelter.

Results and Discussion

Experiment 1. Feeding ecology

The average area of leaf material consumed by P. quercicella was 19.97 ± 0.95 cm2 (median = 18.41 cm2, range = 14.48–34.14 cm2, n = 27). On average, P. cryptolechiella caterpillars consumed 23.79 ± 0.65 cm2 (median = 24.64 cm2, range = 15.08–27.45 cm2, n = 21). When controlling for sex and development time, the average amount of leaf material consumed did not significantly differ between caterpillar species (F1,44 = 0.28, p = 0.5945), though P. cryptolechiella ate an average of 8.5% more leaf material than P. quercicella (Fig. 2A).

Figure 2 Development metrics of Psilocorsis cryptolechiella and P. quercicella caterpillars.

(A) Area of leaf eaten (mean ± S.E.) during larval development, (B) development time (mean ± S.E.) and (C) pupal mass (mean ± S.E.) of Psilocorsis cryptolechiella (n = 8 females, n = 13 males) and P. quercicella caterpillars (n = 13 females, n = 14 males).

Development time, the amount of leaf material eaten, and pupal mass were all positively correlated; caterpillars with longer development times ate more and reached a higher pupal mass. Development time differed significantly between species, with P. cryptolechiella taking an average of 8.41 days, or 35.65% longer to develop than P. quercicella (F1,44 = 36.32, p < 0.0001; Fig. 2B). Development time was also positively associated with area eaten (F1,44 = 6.61, p = 0.0136). Psilocorsis cryptolechiella also achieved a significantly higher pupal mass than P. quercicella, on average weighing 32.90% more than P. quercicella (F1,44 = 28.85, p < 0.0001; Fig. 2C). Females ate more than their male counterparts (F1,44 = 6.01, p = 0.0183; Fig. 2A). However, males and females did not differ in development time (F1,44 = 0.08, p = 0.7846) or in pupal mass (F1,44 = 1.66, p = 0.2037).

Controlling for differences in development time, the two species ate about the same amount of leaf material, with females consuming more than males. Psilocorsis cryptolechiella had a longer development time and a higher pupal mass than P. quercicella, but only ate slightly more than P. quercicella, indicating that it may have higher feeding efficiency. Females and males did not differ significantly in development time or pupal mass, but females ate more leaf material than their male counterparts. This suggests that females metabolize their food differently than males and may have higher resource requirements due to more costly tissues needed for reproduction (i.e., eggs). Research shows that each female P. quercicella moth can lay 200 or more eggs (Lill et al., 2007), which may explain the difference in resource use by males and females. Future studies should explore physiological differences in resource requirements between species and between males and females.

We compared the resource requirements of each leaf-tying species that we studied with their known densities to estimate the amount of plant resources that each species uses in a forest. A previous study by Lill (2004) provided approximate peak densities of each species per 100 leaves at a Missouri field site, with comparable densities to LBRP. Given that the average size of an understory white oak leaf in Maryland forests is 56 cm2 (JT Lill, unpublished data) and using these densities and the average amount of leaf eaten per individual of each species, we estimated the area of leaves consumed by P. quercicella and P. cryptolechiella. This area was doubled to account for two generations of each species in a year. We found that with a density of 1.86 caterpillars per 100 white oak leaves, P. quercicella consumes about 1.33% of the total white oak foliage in a year. Similarly we found that at a density of 1.76–1.98 caterpillars per 100 leaves, P. cryptolechiella consumes 1.50%–1.68% of foliage. A previous study found that Pseudotelphusa quercinigracella consumed 9.55 ± 0.65 cm2 of leaf area throughout development (Lill & Marquis, 2003). Based on these data, we estimated that P. quercinigracella consumes about 2.54% foliage, since their density is estimated to be 7.45 caterpillars per 100 white oak leaves. Due to regional differences in natural densities in each species, with P. quercicella being slightly more common and P. quercinigracella being less common in LBRP, we expect the actual foliage consumed for those two species to be slightly higher and lower than the given estimates, respectively. However, even at a density of 4 P. quercicella per 100 leaves (twice the estimate made by Lill) the area of foliage required to meet the needs of this species is 2.85% of the available foliage. This is notable because several publications by the US Department of Agriculture Forest Service and previous scientific studies have characterized leaf-tying caterpillars as defoliators of oak trees, labeling them as “pests and potential problems”, “damaging agents”, and “threats” and associating them with the invasive Gypsy moth (Lymantria dispar; Cochaux, 1969; Rogers, 1990; USDA NRCS, 2002). It is a stretch to consider consuming less than 3% of the total white oak foliage as significant defoliation.

Experiment 2. Area of natural ties

We collected 158 natural ties throughout the summer at LBRP. The average size of natural ties was 15.51 ± 0.99 cm2 (median = 12.26 cm2, range = 1.23–81.28 cm2). Given the mean leaf area required to sustain the most common leaf-tying caterpillars (21.6 cm2), the area of leaf overlap required to sustain one caterpillar throughout development is about 11 cm2, since each natural tie has two leaves available for consumption. Almost half (46.2%) of the natural ties measured were too small to accommodate an average caterpillar throughout the larval phase and 76.6% were too small to sustain two caterpillars (Fig. 3). This reveals that many caterpillars may be forced to leave their shelter due to a lack of resources within the tie. Caterpillars that inhabit ties that are too small to meet their resource requirements will need to reside in multiple ties, either building new leaf shelters or colonizing existing ties. As the season progresses and existing ties become skeletonized from feeding, there may be a lack of suitable sites where uneaten leaves overlap and can be made into additional ties. As a result, the density of occupants inside leaf ties increases over the course of the season (Sigmon & Lill, 2013), as caterpillars become limited by the availability of sites for building shelters.

Figure 3 Distribution of size of natural ties compared to area required for caterpillars to develop.

Histogram of the number of natural ties collected of different sizes (n = 158). The area of overlap between leaves was doubled to represent the area available for caterpillars to consume. The boxplot aligned with the distribution represents the area of leaf consumed by two species of leaf-tying caterpillars throughout development (cm2).

Leaf-tying caterpillars found in the ties included Psilocorsis quercicella, P. cryptolechiella, P. reflexella (Oecophoridae), Pseudotelphusa quercinigracella (Gelechiidae), Arogalea cristifasciella (Gelechiidae), Morrisonia confusa (Noctuidae), Pococera expandens (Pyralidae), Antaeotricha schlaegeri, A. humilis (Oecophoridae), and Anclis divisana (Tortricidae). The identity of the original builder of the tie influenced the size of natural ties; those built by P. quercicella were 53.4% larger than those built by Pseudotelphusa quercinigracella (F3,70 = 4.56, p = 0.0114; Fig. 4A). Previous studies show that on average P. quercinigracella consume 9.55 cm2 ± 0.65 cm2 leaf material (Lill & Marquis, 2003), just half of what is required to sustain P. quercicella or P. cryptolechiella throughout larval development. The average size of natural ties built by P. quercinigracella was 11.01 cm2 ± 1.34 cm2, which provides enough resource to sustain one of these caterpillars. Therefore P. quercinigracella that are the sole occupant of a tie may not need to abandon ties as often as other species. This may point to a variation in engineering effects between the species, since P. quercinigracella may not build as many microhabitats as other leaf-tiers. Female oviposition choice may also play a role in the size of ties built by different species. Psilocorsis quercicella moths are known to oviposit eggs in groups of 8–12, rather than singly as P. quercinigracella does (M Sliwinski, E Sigmon, personal observations). Females that lay more eggs at a time may choose oviposition sites where larger ties can be built. Future research could be directed towards examining the role of various species in movement and establishing additional shelters, to compare the relative engineering effects of different species of leaf-tiers.

Figure 4 Area of natural tie by (A) the species that built the tie and (B) the number of caterpillars found in the tie.

Relationship between (A) the species that built a natural leaf tie (Psilocorsis quercicella, P. cryptolechiella, P. reflexella, Pseudotelphusa quercinigracella) and the area of the tie (mean ± S.E.), and (B) the number of caterpillars found in a natural tie and the area of tie (mean ± S.E.). Numbers at bottom of bars represent sample sizes. Bars not sharing the same letter are significantly different according to Tukey’s HSD test.

Nearly a third (29.75%) of the natural ties were found empty, indicating that the original tie-builder may have left or been forced out of the shelter. Many of the ties (13.92%) contained two or more leaf-tier occupants, with up to eight caterpillars in one tie, and an average of 1.35 occupants, when excluding ties found empty. Natural ties with three or more occupants were three times larger than those with 0–2 occupants, but ties with 0, 1 or 2 occupants did not differ in size (F3,157 = 16.10, p < 0.0001; Fig. 4B). While this research shows that a sizeable proportion of ties in the wild are too small for one caterpillar, the issue is further complicated by the presence of multiple caterpillars in one tie, and therefore the division of limited resources. Three quarters of the natural ties were too small to support two caterpillars throughout development. Of the natural ties that contained two caterpillars, approximately the same proportion (71.43%) was too small to support two caterpillars throughout development. Some species of leaf tiers have been observed to pull additional leaves into their ties when food resources become low, such as P. expandens that is frequently found in high densities within ties. However, most leaf tiers very rarely add leaves to their shelters as it is uncommon for leaves to be close enough for a single caterpillar to pull it towards an existing tie.

In sum, when comparing the quantity of resources available to caterpillars in leaf ties found in the field to the resource requirements of common species of caterpillars derived from feeding assays, almost half of the ties built appear to be too small for individual P. quercicella and P. cryptolechiella caterpillars to complete development. This alone would force caterpillars to abandon ties in search of more food, creating additional microhabitats or colonizing existing leaf ties in the process. Three quarters of the ties were too small for two caterpillars, even though many ties contain multiple occupants, and caterpillars that are oviposited together may experience densities of 10–20 caterpillars in one tie. High densities of caterpillars further limit resources within a tie, forcing some caterpillars to abandon their leaf tie and build new ones, which increases their overall engineering effect.

Experiment 3. Density effects on caterpillar movement

Caterpillars in density treatments of 2 or 4 were more likely to leave their tie, left sooner, and initially left at a faster rate than those in density treatment 1. In ties with 2 or 4 caterpillars, the first caterpillar left the tie significantly sooner than solitary caterpillars (F2,54 = 14.17, p < 0.0001). There was a marginally significant difference between species in the census period when the first caterpillar left the tie (F1,54 = 3.82, p = 0.0560), with the first P. cryptolechiella/reflexella leaving the tie later than P. quercicella. In ties with 1 caterpillar, the occupant remained in the tie throughout the entire experiment in 25% of the trials; in 60% of the trials, the occupant did not leave until 4 days after the start of the experiment. In 65% of trials with 4 caterpillars and in 50% of trials with 2 caterpillars, an occupant left within one day from the start of the experiment. In 90% of trials with 4 caterpillars and in 75% of trials with 2 caterpillars, an occupant left within the first three days of the experiment. Although many caterpillars that were in density treatment 1 remained in the tie throughout the experiment, 10% of these caterpillars left the tie on the first day, even though they were the sole occupants of the tie. None of the experimental ties had more than 50% of the leaf surface consumed during the experiment, so food limitation was not likely to be a significant factor in leaf tie abandonment in this study. Clearly there are other factors causing caterpillars to abandon their tie in the absence of interactions with other occupants and with sufficient resources available for consumption. Further studies are needed to explore this behavior.

During the course of the experiment, we noticed that several caterpillars were pre-pupal in the later census periods, during which caterpillars typically crawl down from the tree to pupate in the soil. Due to the possibility of caterpillars leaving the tie because of developmental cues, and not as a response to occupant density or pressure on resources, we conducted two ANOVAs. We analyzed effects of density on the proportion of caterpillars remaining in the tie both ending at the fifth census period (Day 3 am) and for the duration of the entire experiment (6 days). The propensity for a caterpillar to leave or remain in the tie was strongly associated with multiple ecological factors (Table 1). One of the main factors is the density of occupants within a tie. Caterpillars in ties as the only occupant were more likely to remain in the tie throughout the experiment than caterpillars in ties of two or four occupants (Table 1; Fig. 5). Species also differed in that a higher proportion of P. cryptolechiella/P. reflexella remained in the tie than P. quercicella (Table 1; Fig. 5). The census period also showed a strong association with the proportion of caterpillars remaining; as time progressed in the experiment, fewer and fewer caterpillars remained in their initial ties (Table 1; Fig. 5).

Figure 5 The proportion of caterpillars remaining in a tie for each density treatment.

The proportion of caterpillars remaining in a tie (mean ± S.E.) at each census period for three density treatments (1, 2, or 4 caterpillars) for (A) Psilocorsis cryptolechiella/reflexella and (B) P. quercicella. A higher proportion of P. cryptolechiella remained in the ties than P. quercicella (F1,648 = 19.21, p < 0.0001). Also, a higher proportion of caterpillars that were the sole occupant remained in the tie than those in density treatments of 2 and 4 caterpillars (F2,648 = 18.31, p < 0.0001).

Table 1 ANOVA results for the proportion of caterpillars remaining in a tie.

ANOVA results for the proportion of caterpillars remaining in a tie occupied by 1, 2, or 4 caterpillars (Density) of either P. cryptolechiella or P. quercicella (Species) censused twice a day for 6 days (Census). The top half of the table refers to results for the entire duration of the experiment (F11,648 = 40.97, p < 0.0001), while the second half of the table refers to results for movement within the first five census periods (F11,288 = 19.18, p < 0.0001). Significant results are marked with *.

Variable	Degrees of freedom	F statistic	P value	
Duration of entire experiment (6 days)				
Density	2	18.31	<0.0001 ***	
Species	1	19.21	<0.0001 ***	
Census	1	378.27	<0.0001 ***	
Density × Species	2	3.34	0.0360 *	
Density × Census	2	0.28	0.7584	
Species × Census	1	4.39	0.0365 *	
Density × Species × Census	2	2.45	0.0870	
				
First 5 census periods (3 days)				
Density	2	12.27	<0.0001 ***	
Species	1	4.00	0.0463 *	
Census	1	80.93	<0.0001 ***	
Density × Species	2	0.24	0.7843	
Density × Census	2	4.76	0.0092 **	
Species × Census	1	1.31	0.2533	
Density × Species × Census	2	1.13	0.3250	

The most striking difference between the ANOVA analyses for the two time spans is the rate at which caterpillars in different density treatments left the ties (density × census interaction; Table 1). Caterpillars that were the sole occupant of the tie left at a slower rate in the first three days than those in density treatments 2 and 4, but left at a similar rate when considered across the entire 6 days (Fig. 5). At least one caterpillar per tie in the higher density treatments left in the first few days while solitary caterpillars remained in their ties until near the end of the experiment. This indicates that caterpillars are very likely to abandon a leaf tie with multiple occupants, but behavioral interactions and food limitation are not the only reasons caterpillars abandon their ties.

The two species studied also differed in their propensity to leave the shelter, particularly at the end of the experiment. The rate at which caterpillars left the tie differed between species (species by census interaction) for the whole experiment, but not for the first 3 days (Table 1). For the whole experiment, P. quercicella left the ties more quickly than P. cryptolechiella (Fig. 5). In addition, for the entirety of the experiment there was a significant density by species interaction (Table 1; Fig. 5). A higher proportion of P. cryptolechiella caterpillars in density treatment 4 remained than those in density treatment 2, whereas the opposite was true for P. quercicella. One explanation for these species differences may be the different shapes of the shelters built by these species. Psilocorsis quercicella builds leaf ties in which the area that is sealed with silk and frass is in a large circle, whereas P. cryptolechiella typically build a small narrow tunnel of frass that they remain in when not feeding (Figs. 1B and 1C). This may allow for more P. cryptolechiella shelters than P. quercicella shelters to fit in the same area. Psilocorsis quercicella are typically laid in large groups, which may make them more adapted to dispersing from the leaf tie they are originally oviposited in. Another possible reason is that a study showed that P. cryptolechiella caterpillars are more aggressive than P. quercicella (E Sigmon, unpublished data). Perhaps P. quercicella are more likely to disperse to avoid conflict, while P. cryptolechiella effectively establish and maintain territories within the tie. Unfortunately, we were unable to conduct interspecific trials due to a limited number of caterpillars available, but this would be an interesting area for further research.

Caterpillars were also more likely to leave overnight than during the day (p = 0.0349). This was particularly apparent by the third day, after which only 4 caterpillars, all of which were P. quercicella in density treatment 4, left a tie during the day. Caterpillars likely move overnight in order to avoid detection by visually oriented predators such as birds and wasps. We were able to relocate 21 caterpillars that built new shelters, which were 27.62 cm ± 3.77 cm (median = 20 cm, range = 5–70 cm) away from the original shelter. Of the caterpillars that moved and were relocated, 38.09% built new shelters, while 61.91% caterpillars occupied pre-existing “bait” ties placed at 20 cm, 40 cm or 60 cm. It is notable that more than a third of the caterpillars built additional leaf ties when uneaten, unoccupied ties were available. When not limited by site availability, caterpillars build new leaf ties nearby the tie they have abandoned. However, in the field sites, the ability to build ties may be limited by plant architecture and the presence of other caterpillars. Further research should address the role of site availability, along with occupant density interactions and resource requirements in the dispersal of leaf-tying caterpillars.

This experiment demonstrates that ties containing more than one caterpillar will most likely be abandoned by at least one occupant. However, solitary caterpillars were also observed to leave their ties, even with an abundance of food available to them in the tie. These findings suggest that it is very likely that most caterpillars will build multiple ties in order to complete development, and thus increase the heterogeneity of the habitat for secondary occupants. Additional research is needed to explore how many leaf ties a single caterpillar builds or resides in throughout its larval lifespan by following individual caterpillars. This would provide insight into precisely how many ties each caterpillar creates, and thus the magnitude of the engineering effect. Also, a study of how many arthropods inhabit a single tie throughout its existence would allow for an estimation of the magnitude of the engineering effect of leaf tiers and the effect of caterpillars creating multiple ties. By creating multiple ties during a lifetime, leaf-tying caterpillars likely have an even greater engineering effect than previously understood.

Experiment 4. Searching behavior

Caterpillars are very vulnerable while outside of a leaf tie. Of the 29 caterpillars observed, 5 died during the observation period. Wasps captured two caterpillars and three caterpillars fell off the tree to the leaf litter where we presumed them dead. Though the caterpillars likely survived the fall, there are many predators in the leaf litter and we believe there is a very small chance of the caterpillar finding and climbing the correct tree species to build a new shelter. Seven other caterpillars fell off the tree but remained attached by silk and were able to get back onto the branch. These observations were not done on windy days, so the caterpillars were not knocked off the tree, but simply fell while crawling. The attacks by wasps, which occurred after only 9 and 13 min of searching, support our assumption that caterpillars abandon ties more frequently overnight to avoid visually oriented predators. Searching for a site to build a shelter appears to be a risky behavior that we would expect caterpillars to avoid. Thus it is perplexing that caterpillars frequently leave their shelter even when they are the sole occupants and food resources are sufficient, as in the solitary caterpillars in Experiment 3.

Leaf-tying caterpillars travel far and spend a long time searching for a site to build a leaf tie. Seven caterpillars did not move for over an hour and the observations were ended due to time constraints. For the 17 caterpillars that built a shelter, they took 28 ± 7 min (range: 6–140 min) to find a place to build a shelter and rested for 11 ± 7 min (range: 0–120 min) of this time. The caterpillars traveled 110 ± 20 cm (range: 13–314 cm) while searching for a shelter site. These caterpillars are ∼1 cm in length, so this is a relatively long distance. Caterpillars explored 3.8 ± 0.6 (range: 1–9) leaves and revisited 1.7 ± 0.5 (range: 0–7) leaves while searching for a place to build a shelter. Caterpillars have very poor eyesight and were observed coming within a few centimeters of a pair of overlapping leaves and seemed to not recognize the suitable shelter site. Thus we speculate that they find suitable sites by chance rather than meticulous searching. Two caterpillars were observed feeding while searching for a site to build a shelter. Leaf-tiers were previously thought to never feed outside of their shelter, but this shows that they will feed without the protection of a shelter when necessary. Searching for a site to build a shelter is a costly activity, both in time and energy, which increases the risk of predation and dislodgement from the tree.

Conclusions

Leaf-tying caterpillars likely abandon their ties due to a combination of factors including a lack of resources and density-related interactions with other leaf tie occupants. Almost half of the natural ties found were too small to support the development of an average caterpillar and three quarters of the ties were too small for two caterpillars. Caterpillars that inhabit ties that are too small to meet their resource requirements will need to reside in multiple ties, either building new leaf shelters or colonizing existing ties. This issue is further complicated by the presence of multiple caterpillars in one tie, and therefore the division of limited resources. Occupant density had strong effects on caterpillar movement, with caterpillars in ties with more than one occupant leaving the ties sooner and at a faster rate than those that were the sole occupants of ties. Moreover, P. quercicella left the ties at a faster rate than P. cryptolechiella, possibly due to their larger requirements for territorial space or their more submissive behaviors. Caterpillars were more likely to leave the tie overnight than during the day, most likely in an effort to avoid detection by predators, which are a major threat when caterpillars search for a site to build a new leaf tie. Caterpillars spend on average a half hour and travel 1 m to find a new site to build a leaf tie, but this is very variable. We conclude that it is likely that most caterpillars will build multiple ties in their larval lifetime, and in doing so, create microhabitats that foster greater biodiversity and abundance of the arthropod community.

Many thanks go to John T. Lill for guidance in this research. We also thank Mariana Abarca, Katherine Costantini, Arjun Aswathi, and Luke Fey for their assistance in collecting and rearing caterpillars and in data collection. We would like to thank reviewers for their helpful comments.

Additional Information and Declarations

Competing Interests

Author Contributions

Field Study Permissions

The authors declare there are no competing interests.

Michelle Sliwinski and Elisha Sigmon conceived and designed the experiments, performed the experiments, analyzed the data, contributed reagents/materials/analysis tools, wrote the paper.

The following information was supplied relating to ethical approvals (i.e., approving body and any reference numbers):

A special use permit was approved by Montgomery Parks.

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
