# Peer review of "Why do leaf-tying caterpillars abandon their leaf ties?"

_PeerJ, doi:10.7717/peerj.173_

## Round 0.1 · original submission · Minor Revisions

Both reviewers agreed that this is an interesting, well written manuscript that will advance our knowledge of these systems. I agree. Both reviewers also raised a number of specific and relatively straightforward points that you should be able to address in a revision. If you have any questions about their comments, please feel free to get in touch with me.

Reviewer 1 ·

Basic reporting

In this manuscript by Sliwinski and Sigmon, the authors present a detail experimental study on leaf-tying caterpillars exploring the propensity for caterpillars to leave their ties. Specifically, they look at the size of ties needed for caterpillars to reach maturity, how often caterpillars leave ties in the presence or absence of other individuals within the ties, and their fate after dispersing from the original ties (i.e. colonizing another tie, building a new one, or potentially being eaten). From the study, its clear the authors have a detailed knowledge of the natural history of this system and the two focal species used in the study. I found the manuscript to be well written and I liked the combination of Results and Discussion. Taken together, this is an interesting behavioral study of leaf-tiers, which have become near a model system for the study of the ecological role of ecosystem engineering.

Experimental design

In the feeding experiment (Experiment 1), the authors could have looked at caterpillar growth rates in the presence and absence of other individuals, which could have supported their argument for leaving crowded shelters, if they would have found fitness consequences of competition for shared shelter. Something to keep in mind for future studies.

In experiment 3, are caterpillars typically competing with members of the same species for a tie? If so, it would make sense to combine individuals of the same species together. If they are found with other species, then it’s unclear why the authors didn’t partition the experiment to intra- and inter-specific density manipulations. Perhaps some simple clarification is all that is needed here.

Experiment 4 is probably the weakest quantitative aspect of the paper. But, I appreciate the inclusion of the observational data for future studies interested in predation and the protection of caterpillars through leaf tying.

Validity of the findings

No Comments

Additional comments

I found the tendency for caterpillars to leave the leaf-ties despite no other caterpillars present and food resources being adequate, as well as the difference in the two species, to be the most interesting aspect of the paper. I believe the manuscript would be strengthened if the authors could provide additional links from their results to the community consequences. For example, I believe they could estimate how many additional ties are potentially created by caterpillars leaving ties, as well as add some back-of-the envelope calculates to the secondary effects on the inquiline community. These estimates could be based on prior studies by Lill and Marquis on to the average number of secondary user species within these ties.

Figure 1 and 6 could be combined.

Figure 5, provide lowercase lettering for significant pairwise differences within each time period.

·

Basic reporting

Line 261–“60% of trials” is misspelled as “trails”

Experimental design

Exp 1–Were there any assessment of potential differences between field-collected and lab-reared caterpillars in terms of the performance measures? What’s the proportion of these two types of caterpillars used in the analysis? If some individuals came from the same lab colony, or from the same brood of eggs in the field, was the non-independence accounted for in the analysis? For the caterpillars kept in deli containers, the leaves were replaced weekly, which means that they had to construct new ties every week. This would seem to generate extra energy requirement for the caterpillars and might result in over-estimation of their feeding amount in comparison to natural conditions.
Exp 3–How were the caterpillars “relocated” after they have moved and the distances they moved determined? Were they individually marked? When it says “proportion of caterpillars remaining in a tie”, does it mean the proportion out of the 1, 2, or 4 caterpillars in each tie? If so, won’t the proportion for the single caterpillar treatment be just 0 or 1? Or are there multiple replicates for each treatment per trial? I don’t see this information in the Methods. And it would seem sensible to quantify the skeletonization of the experimental ties at the end and see if that correlates with the retention rate of caterpillars, especially since the authors mentioned in the Introduction that food resource depletion might be a factor in driving them out. Also, given the potential difference in aggression level between the two species of caterpillars and the commonness of tie sharing by multiple species of leaf tiers, it might be interesting to include a treatment to test intra- vs. inter-specific competition in driving caterpillars out of a leaf tie. However, I realize it is unfeasible to append such a supplement to the current study. So this is more of a suggestion for something to consider for future studies.

Validity of the findings

Exp 3–If the retention rates are comparable between the low and high density treatments during the later stage of the experiment, it would seem to suggest that the developmental cue of pre-pupal caterpillars has played a role as the authors have suggested. But this doesn’t get mentioned again in the discussion. And I don’t quite understand the statement “while solitary caterpillars remained in their ties until near the end of the experiment” (line 285). Does this just mean that a higher proportion of solitary caterpillars remained, rather than all of them remained?
Exp 4–Much of the conclusions in this section appears to be speculative, so probably should be better identified as the review criteria suggest.

Additional comments

This is a well written manuscript presenting a thorough study on the behavior of shelter abandonment by leaf-tying caterpillars. The authors conducted both manipulative experiment to assess the effect of caterpillar density on abandonment rate and observational study to document the caterpillars’ behavior in searching for new ties or building sites. The findings will contribute to our understanding of the causes of this behavior, and help better predict the potential effect on the arthropod community that utilizes these leaf ties.

---

## Round 0.2 · accepted · Accept

Nice work in dealing with the revisions. Your clarifications and additional work have improved an already nice paper.